# Sex Disparity in the Nutrition-Related Determinants of Mild Cognitive Impairment: A Case–Control Study

**DOI:** 10.3390/nu17020248

**Published:** 2025-01-10

**Authors:** Mengjie He, Danting Su, Ronghua Zhang, Peiwei Xu, Dan Han, Lichun Huang, Yan Zou

**Affiliations:** Department of Nutrition and Food Safety, Zhejiang Provincial Center for Disease Control and Prevention, Hangzhou 310051, China; mjhe@cdc.zj.cn (M.H.); dtsu@cdc.zj.cn (D.S.); rhzhang@cdc.zj.cn (R.Z.); pwxu@cdc.zj.cn (P.X.); dhan@cdc.zj.cn (D.H.); lchuang@cdc.zj.cn (L.H.)

**Keywords:** mild cognitive impairment, elder people, nutrition, sex, determinant

## Abstract

Background/Objectives: Sex differences in nutrition-related determinants of mild cognitive impairment (MCI) exist among the elderly. This study aimed to explore sex-specific influencing factors of MCI. Methods: A case–control study was conducted in 2020 involving 1086 elderly people aged 55 years and above from four sites in Zhejiang Province, China. Data on demographics, cognitive assessment, depression scale, daily food intake, and physical examinations were collected. The assessment of plant-based diet patterns depended on an overall plant-based diet index (PDI), a healthful plant-based diet index (hPDI), and an unhealthful plant-based diet index (uPDI). Multivariate logistic regression models were employed to assess the determinants of MCI in females and males. Results: Among 571 females, 141 (24.7%) had MCI, and 126 (24.5%) had MCI among 514 male participants. In females, the multivariate analysis revealed that being unmarried/divorced/widowed (OR = 1.95, 95% CI: 1.10–3.45), having depression (OR = 6.06, 95% CI: 1.87–19.66), and having a uPDI score ≥ 55 (OR = 2.41, 95% CI: 1.50–3.89) were associated with a significantly elevated risk of MCI. Conversely, a cereal consumption of ≥300 g/d (OR = 0.32, 95% CI: 0.19–0.53) was linked to a significantly reduced risk. In males, vegetable consumption ≥ 150 g/d (OR = 0.39, 95% CI: 0.23–0.66), vegetable oil consumption ≥ 22 g/d (OR = 0.502, 95% CI: 0.307–0.820), and cereal consumption ≥ 300 g/d (OR = 0.44, 95% CI: 0.27–0.71) were associated with a lower MCI risk. Meanwhile, rural residence (OR = 1.90, 95% CI: 1.12–3.25) and advanced age, especially 75 years old and above (OR = 4.71, 95% CI: 2.44–9.12), were also risk factors in males. Notably, the Restricted Cubic Spline (RCS) model showed that females with a uPDI score < 55 had a lower prevalence of MCI, while those with a score ≥ 55 faced a higher risk. Conclusions: This study indicates potential sex disparities in the risk factors for MCI. Future research should prospectively establish causal relationships. Additionally, precise intervention strategies are urgently needed.

## 1. Introduction

The issue of global aging is becoming increasingly severe. As World Population Prospects 2024 reported, 10.5% of the global population was aged 65 years and older [1]. According to the statistical Bulletin of the People’s Republic of China on National Economic and Social Development published in 2024, China transitioned into an aged society as the proportion of its elderly population (aged 65 years and above) surpassed 15.4% [2]. The health risks entailed by aging can impose substantial burdens on families, health systems, and society. Alzheimer’s disease (AD) accounts for 60–70% of all dementia cases and ranks as the primary cause of death among individuals aged 75 years and above [3,4]. The incidence rates of AD in people aged 65 and 85 years old are approximately 5% and over 30%, respectively [5]. Globally, it is estimated that there are around 55 million AD patients, with 10 million of them in China [4,5]. It is projected that the global number of AD patients will reach approximately 130 million by 2050 [5]. To date, there is no effective cure available for delaying the progression of AD. Therefore, primary prevention of AD is of crucial importance. Mild cognitive impairment (MCI) represents an intermediate stage that lies between the typical cognitive decline associated with normal aging and dementia, ten percent to 20% of which progress to manifest dementia within 12 months of diagnosis [6]. Exploring the influencing factors of MCI, identifying high-risk groups, providing early warnings, and paying close attention are the keys to preventing AD.

Nutrition plays a pivotal and indispensable role in the senescence process of the brain [7]. Therefore, it is of utmost importance to explore the evidence regarding the association between nutrition-related factors and the risk of MCI in the elderly. This exploration aims to redirect the focus towards preventive strategies for this pre-dementia stage of AD. The associations between dietary and cognitive functions have continuously captured the attention of the general public. Particular attention has been dedicated to fresh vegetables and fruits since they serve as outstanding sources of antioxidant nutrients, including vitamin C, vitamin E, and carotenoids. Moreover, the consumption of fish and nuts has been given emphasis due to their abundance of unsaturated fatty acids, which have been empirically proven to exhibit anti-inflammatory benefits [8,9,10,11]. A stronger adherence to a Mediterranean-style dietary pattern correlates with a diminished risk of contracting Alzheimer’s disease and a decelerated rate of cognitive decline as one ages [12]. Several studies have indicated that a higher intake of long-chain omega-3 polyunsaturated fatty acids (LC ω3 PUFAs) may serve as a preventive approach against Alzheimer’s disease. Notably, this preventive effect is most pronounced when dietary LC ω3 PUFAs are ingested prior to, or during the early phases of, cognitive decline [13]. In addition to the research reports on the association between food nutrient intake and MCI, there are also research reports on the relationship between human nutritional status and MCI. A systematic review and meta-analysis reported that the prevalence of MCI is relatively high in patients with sarcopenia, and sarcopenia may be a risk factor for MCI [14]. Although some research reports have emerged regarding the nutrition-related influencing factors of MCI, the consistency within the available evidence remains inadequate. Moreover, few studies have been dedicated to comparing the disparities in the nutrition-related influencing factors of MCI between males and females, which is due to the fact that women and men possess distinct physiological states, endocrine systems, and, most notably, dietary habits. These differences also manifest in the prevalence of common medical conditions, especially those related to the cognitive domain. Previous reports have already indicated that women and men face varying risks of cognitive impairment [15,16,17,18]. However, there remain inconclusive differences in nutrition-related predictive factors or preventable influencing factors of MCI between different sexes. This study aimed to explore the nutrition-related influencing factors of MCI between males and females by using the method of case–control study, thereby achieving precise prevention of MCI.

## 2. Materials and Methods

### 2.1. Study Design

Data in the present study were obtained from the follow-up of Community-based Cohort Study on Nervous System Diseases. This study concentrated on potential factors correlated with the risks of three neurological disorders: epilepsy in subjects over 1 year of age, and AD and Parkinson’s disease among the population aged 55 years and older [19]. Present study targeted subjects recruited in the cohort of AD. In Zhejiang Province, the field investigation and physical examination were conducted in September 2020. The protocol of this project was reviewed and approved by the Medical Ethics Committee of the National Institute for Nutrition and Health, Chinese Center for Disease Control and Prevention (Approval No. 2017020, 6 November 2017). Written informed consent was obtained from each participant. Specifically, after the research protocols were meticulously explained to them, the participants provided their written informed consent.

### 2.2. Sampling Method and Study Population

Participants without such diagnosed diseases were recruited using a multistage stratified random sampling approach at baseline. In Zhejiang Province, there were four study sites, including two cities and two counties, that represented both urban and rural areas across the province. Subsequently, urban and suburban neighborhoods in the selected cities, together with townships and villages in the counties, were randomly sampled. Ultimately, all members of a randomly selected household who met the inclusion and exclusion criteria for any of the three nervous system diseases were interviewed [19].

In this study, the inclusion criteria for eligible samples were as follows: (1) aged 55 years or older; (2) being part of the resident population in the sampled community; and (3) free from comorbid conditions that could interfere with the assessment, such as congenital or acquired mental retardation, and uncorrectable visual or hearing abnormalities [19]. At the beginning of the study, 2790 subjects were included in the survey. Subjects with complete data regarding basic and health, cognitive examination, psychological evaluation, physical examinations, and a survey of basic daily living abilities were selected to participate in the present study (*n* = 1103). Based on the definition of MCI, subjects were excluded if they were unable to carry out basic activities of daily living, including eating, dressing, bathing, using the toilet, grooming, transferring between the bed and chair, walking across a room, and maintaining urinary or fecal continence (*n* = 17). Ultimately, a total of 1086 participants were included in the analysis.

### 2.3. Data Collection

Investigators were required to hold a degree in medicine or public health. They underwent two rounds of training, which were, respectively, conducted by national experts and provincial professionals. Those who successfully passed the qualification test were assigned to collect information through questionnaires and physical examinations. Questionnaire survey was used to collect basic and health-related data regarding age, educational level, occupation, marriage, cognitive assessment, depression scale, and dietary information, including daily consumption of 81 categories of food through Food Frequency Questionnaire (FFQ). Physical examinations, including height, weight, waist circumference, grip strength, 6-metre gait speed, and body composition, were carried out by trained health workers from the local community health center.

### 2.4. Cognitive Function Assessment

In the field survey, both the Mini-Mental State Examination (MMSE) and the Montreal Cognitive Assessment (MoCA) were used simultaneously to screen for MCI. However, the MoCA scale was employed to assess MCI. The reason was that our research team had previously conducted a study comparing the MMSE with the MoCA for MCI screening in the Chinese middle-aged and elderly population. The findings suggested that MoCA is a superior measure of cognitive function. It lacks a ceiling effect and is highly effective at detecting cognitive heterogeneity [20]. Therefore, in this study, the cognitive function of participants was evaluated using only the MoCA. This tool has demonstrated validity and reliability among the Chinese population, as it takes into account cultural and linguistic differences [21,22]. The specific cultural and linguistic adaptations made to the original English version to develop the MoCA Beijing version were also described in detail [23]. The MoCA evaluations were conducted strictly face-to-face by trained investigators in strict accordance with the guidelines and protocols. These assessments were completed within 5–10 min for some parts and 10–15 min for others. The MoCA consists of 52 items, with scores from 32 of these items being tallied to yield a total score ranging from 0 to 30 points, which is positively correlated with global cognitive function [24]. The criteria for MCI were based on Chinese MoCA norms [22]: an illiterate individual was considered to have MCI if their total MoCA score was ≤13; for those with 1–6 years of education, the cutoff was ≤19; and for individuals with 7 or more years of education, the cutoff was ≤24.

### 2.5. Plant-Based Diet Indices and Dietary Assessment

Extract 18 types of plant-based dietary-related foods were extracted from the 81 categories of foods in FFQ. The assessment of healthful and unhealthful plant-based diets depended on an overall plant-based diet index (PDI), a healthful plant-based diet index (hPDI), and an unhealthful plant-based diet index (uPDI) [25]. Food groups were classified into quintiles and assigned either positive or reverse scores. In the case of positive scores, participants falling above the highest quintile of a particular food group were allocated a score of 5, and this score decreased incrementally down to 1 for those below the lowest quintile. Conversely, for reverse scores, the scoring pattern was inverted. To construct the PDI, plant food groups were assigned positive scores, while animal food groups received reverse scores. Regarding the hPDI, positive scores were given to healthy plant food groups, and reverse scores were assigned to less healthy plant food groups and animal food groups. Finally, for the uPDI, positive scores were attributed to less healthy plant food groups and reverse scores were given to healthy plant food groups and animal food groups, as shown in Appendix A. The scores of the 18 food groups were then aggregated to obtain the respective indices. The grouping cut-off values of cereals, vegetables, fruits, soybean and their products, vegetable oils, and nuts were determined through the RCS model. Strong tea was grouped according to whether it had been drunk. Strong tea was defined as the ratio of tea leaves to water at 1:50.

### 2.6. Assessment of Covariates

Geriatric Depression Scale (GDS) was used in this study to screen depressive symptoms. We adopted the cutoff value of 11, which was recommended by the developer and utilized in most studies, for screening depressive symptoms [26]. Central obesity was defined as a waist circumference of ≥90 cm for men and ≥85 cm for women, in accordance with the criteria for adult weight in China [27]. Body mass index (BMI) was computed by dividing weight (in kilograms) by the square of height (in meters), and the subjects were then classified into four groups [28], including lean, normal, overweight, and obesity. Muscle mass was measured via bioelectrical impedance analysis (BIA) using a TANITA 601 device (Tanita Corporation of America, Inc, Arlington Heights, IL, USA). Appendicular skeletal muscle mass (ASM) is defined as the mass of skeletal muscle located in the arms and legs. To obtain the skeletal muscle mass index (SMI) (expressed in kg/m^2^), ASM is standardized by height, which is expressed in meters squared. Muscle strength was evaluated through handgrip strength, measured with a digital grip strength dynamometer (CAMRY EH 101, Beijing, China). Measurements were taken for each hand, and only the highest value was recorded. Physical performance was appraised by the 6 min gait speed, with the average of two tests being used. Sarcopenia, low muscle mass (men: SMI < 7.0 kg/m^2^, women: SMI < 5.7 kg/m^2^), low muscle strength (men: handgrip strength < 28 kg, women: handgrip strength < 18 kg), and low performance (6 min gait speed < 1.0 m/s) were defined based on the recommendations of the Asian Working Group for Sarcopenia 2019 (AWGS2019) consensus [29].

### 2.7. Statistical Analyses

Mean and standard deviations (mean ± SD) were employed to describe continuous variables with a normal distribution, while median and quartiles [median (quartiles)] were utilized for variables presenting a skewed distribution. Frequencies and percentages (%) were reported for categorical variables. Categorical variables were compared using the chi-square test. A series of multiple logistic regression models were established to evaluate the odds ratio (OR) and the corresponding 95% confidence interval (CI) of potential influencing factors, including age, educational level, occupation, marriage, depressive symptoms, daily consumption of coffee, nuts, strong tea, chocolate, overweight/obesity, central obesity, sarcopenia and incidence of MCI. Additionally, the figures were plotted based on Restricted Cubic Spline (RCS) Model. All statistical analyses were performed by using the program SAS version 9.1. *p* less than 0.05 was considered statistically significant.

## 3. Results

### 3.1. Basic Information

A total of 1086 elderly people aged 55 and above, among whom there were 514 men and 572 women, were enrolled in the current study. The main characteristics of the participants are presented in Table 1. Overall, the prevalence of MCI in women was 24.7%, and in men was 24.5%.

### 3.2. Nutrition-Related Determinants of MCI in Female and Male

Based on the results of the existing research evidence and univariate analysis, region, age, educational level, marriage, intakes of cereals, vegetables, fruits, soybean, nuts, and vegetable oils, uPDI, hPDI, and sarcopenia were included in the multivariate analysis. Multivariate analysis showed that among all the elderly population, cereal intake was associated with MCI in Table 2 and Table 3. Meanwhile, depression, unmarried/divorced/widowed, and high uPDI were risk factors for MCI only in female elderly people, while residing in urban and appropriate intakes of vegetables and vegetable oil were protective factors for MCI only in males. The risk of MCI in females was higher with depression (OR = 6.06, 95% CI: 1.87~19.66) or not married (OR = 1.95, 95% CI: 1.10~3.45). Consumption of cereal ≥ 300 g/d (OR = 0.32, 95% CI: 0.19~0.53) was associated with a lower MCI risk in females. The risk of MCI in males was higher when residing in rural areas (OR = 1.90, 95% CI: 1.12~3.25) and increasing age. Consumption of vegetables ≥ 150 g/d (OR = 0.39, 95% CI: 0.23~0.66), consumption of vegetable oil ≥ 22 g/d (OR = 0.50, 95% CI: 0.31~0.82), consumption of cereal ≥ 300 g/d (OR = 0.44, 95% CI: 0.27~0.71), were linked to a significantly reduced risk in males. Appropriate cereal intake was a nutrition-related protective factor against MCI for both females and males, while high uPDI was to have a risk effect only among females, and appropriate intake of vegetables and vegetable oil was found to have a protective effect only among males.

### 3.3. Analysis of the Association Between Plant-Based Diets and MCI in Female and Male

In the univariate analysis, we found that among women, low PDI (*p* = 0.04), low hPDI (*p* = 0.03), and high uPDI (*p* = 0.0002) were significantly positively associated with MCI. Among men, only high uPDI (*p* = 0.006) was significantly associated with MCI. After adjusting for other factors, multivariate analysis showed that uPDI was a risk factor for MCI in females. RCS was established for the relationship between the uPDI and MCI by adjusting the factors including age, educational level, depression, and cereal intake in Figure 1. Females with a uPDI score < 55 had a lower prevalence of MCI, while those with a score ≥ 55 faced a higher risk. After using 55 as the cut-off value for uPDI, the results of the multivariate analysis showed that uPDI score ≥ 55 (OR = 2.41, 95% CI: 1.50~3.89) was associated with an increased risk of MCI in females.

## 4. Discussion

Considering that there is no cure for AD, it is particularly important to carry out primary prevention and early warning signs for MCI previous to AD. Nutrition, as one of the important influencing factors, has drawn the attention of scholars. In order to achieve precise prevention, it is crucial to explore the risk factors for MCI in men and women separately. However, to date, such research is lacking. This study aimed to explore the nutrition-related influencing factors of MCI between males and females by using the method of case–control study, thereby achieving precise prevention of MCI. In this study, among 571 females, 141 (24.7%) had MCI, and 126 (24.5%) had MCI among 514 male participants. In females, the multivariate analysis revealed that being unmarried/divorced/widowed, having depression, and having a uPDI score ≥ 55 were associated with a significantly elevated risk of MCI. Conversely, a cereal consumption of ≥300 g/d was linked to a significantly reduced risk. In males, vegetable consumption ≥ 150 g/d, vegetable oil consumption ≥ 22 g/d, and cereal consumption ≥ 300 g/d were associated with a lower MCI risk. Meanwhile, rural residence and advanced age, especially 75 years old and above, were also risk factors in males. Notably, the RCS model showed that females with a uPDI score < 55 had a lower prevalence of MCI, while those with a score ≥ 55 faced a higher risk.

Synthesizing the existing research evidence so far, at the same time, considering the preference differences in dietary choices between men and women, the factors to be included in the analysis were selected. Since plant-based foods can provide rich antioxidant nutrients and play a significant role in the aging process of the brain, the plant-based diet pattern as well as some food types related to antioxidant and anti-inflammatory effects were introduced in this study, including vegetable, fruit, nuts, and vegetable oil. Some research reports in European and American regions suggest that caffeine and chocolate are protective factors against AD. A cohort study indicated that a low caffeine intake correlates with a higher risk of developing amnestic symptoms among patients with MCI or AD. Additionally, caffeine intake has been found to be associated with cerebrospinal fluid biomarkers in AD patients [30]. A nested case–control study conducted in Spain suggested that regular consumption of dark chocolate may enhance cognitive function among the elderly population [31]. However, it was found in Appendix A that the number of elderly people consuming coffee and chocolate was too small, so they were not included in the analysis. Strong tea is a beverage that Chinese elderly people are quite fond of, and there is also a certain amount of caffeine in strong tea. Therefore, strong tea was analyzed as an influencing factor in this study. A study conducted in Japan has indicated that a dietary pattern with a relatively high protein intake is correlated with favorable cognitive function among the elderly population in Japan [32]. Therefore, milk, soybean, and meat were analyzed, but only soybeans showed significance in the univariate analysis. Some studies reported [33,34,35] that simple carbohydrates, such as sucrose (comprising glucose and fructose), may exert an immediate impact on cognitive function. In the long run, diets rich in complex carbohydrates, like fruits and cereals, are associated with better cognitive function and a reduced risk of dementia. Therefore, cereal and sweet foods were included in this study. However, the consumption of sweet food among elderly people is very low, as shown in Appendix A, and the results were not significant. Meta-analysis reported that the prevalence of MCI is relatively high in patients with sarcopenia, and sarcopenia may be a risk factor for MCI [14]. Overall, plant-based diet indices (PDI, hPDI, and uPDI), cereal, vegetables, fruit, soybean, vegetable oil, nuts, strong tea, and sarcopenia were analyzed in this study. Based on the risk factors of MCI reported in previous studies, factors such as region, age, education level, marital status, living alone, occupation, BMI, and depression were included as adjustment factors. It has been reported that loneliness may mediate the association between depression and the quality of life among old populations with MCI.

As this study showed, residing in rural areas and older age, especially aged 75 and above, were risk factors for MCI, which was consistent with most of the current research evidence [36,37]. However, residing in rural areas as a risk of MCI was only observed in males, which needed further exploring the reasons. The educational level, which is a common influencing factor of MCI reported in previous studies, showed no significance in our study. It was most likely related to the fact that the proportion of the elderly with a high educational level included in this study was very low, and the distribution of educational levels was unbalanced. Unmarried/divorced/widowed and depression were reported as risk factors of MCI only in older females. Women with depressive symptoms have a six times greater risk of MCI than women without depressive symptoms. The result was consistent with the previous research report that depression is a risk factor for MCI among women [38], which may be caused by the relatively high incidence of depression among women [39]. However, the OR was a little higher than the depression relative risk (RR) and OR of mild MCI reported in the current cohort and cross-sectional data [38,40,41]. Not being in a marital status may increase the risk of MCI through the mediating effects of loneliness and depression [39,41]. Moreover, findings suggested that among older adults with MCI, being married may be a protective factor, and being unmarried may be a risk factor for experiencing loneliness and subsequent intrusive thoughts [42]. Therefore, marriage and depression were both included in the multivariate analysis to rule out the interference of collinearity, and the results demonstrated that both factors were risk factors for MCI among women. The analysis of the common risk factors for MCI in different sexes, as mentioned above, indicated a relatively high data reliability and was in good consistency with previous studies [16]. There were few reports on the correlation analysis between the plant-based diet pattern and MCI in previous studies. Consequently, this represents a relatively novel research endeavor. A study on the association between plant-based dietary patterns and cognitive function in middle-aged and older residents of China demonstrated that there was a significant association between higher uPDI scores and higher odds of MCI, with Quintile 4 compared with Quintile 1 showing an odds ratio of 2.21 [43], which it was similar to the results of our research. A study on the association between Ultraprocessed Foods and Cognitive Decline also confirmed that an unhealthy diet might pose a potential hazard to cognitive function [44]. Compared with this research, our advantage lies in the fact that the possible influencing factors of MCI that we used for correction are more comprehensive, such as depression. More importantly, we found that in the correlation analysis between the plant-based diet pattern and MCI, the plant-based diet pattern was significant as an influencing factor only in women. Hence, females with an unhealthy plant-based dietary pattern are among the high-risk groups for MCI. Furthermore, it is worth noting that there is an interesting phenomenon that females with a uPDI score < 55 had a lower prevalence of MCI, while those with a score ≥ 55 faced a higher risk. This can provide a basis for dietary guidance and suggestions for the high-risk population of MCI. We found that a low intake of cereals might be a risk factor for MCI in men and women. At present, there are hardly any correlational studies on cereals and MCI, but only on carbohydrates and MCI [33,34]. Research on food types is of greater public health significance compared to nutrients. It can not only provide a basis for the Chinese Dietary Guidelines or the Food Pagoda but also offer references for the dietary intake of elderly people. The influence of the rational consumption of vegetable oils and vegetables on MCI exhibits a disparity in the sexes. Whether this sex difference is caused by different physiological functions between men and women or food preference still requires further investigation. Existing research reported that consuming PFBc in vegetable oil might enhance cognitive function via its anti-inflammatory antioxidant functions [45]. However, the threshold value for the intake of vegetable oil has not been reported. The threshold value of 22 g obtained through the RCS curve in this study showed a significant difference in the risk of MCI. In the research on the association between vegetable intake and MCI, there is relatively little quantitative comparison of vegetable intake, and most of it is in the form of semi-quantitation, such as frequency or quantile distribution [9,46,47]. Therefore, the strength of this study was providing a scientific basis for dietary recommendations for the prevention of MCI.

There were several strengths in our study. Firstly, this study explored the nutrition-related influencing factors of MCI, especially exploring the association between plant-based dietary patterns and MCI. Secondly, this study compared the differences in general influencing factors and nutrition-related influencing factors of MCI between different sexes, and, to date, only a limited number of similar studies have been reported. Finally, the quantitative cut-off value of the effects of uPDI in the plant-based dietary pattern, cereal, vegetables, and vegetable oils on MCI are reported in the study.

However, this study has the following limitations, and caution should be exercised in interpreting and extrapolating the results. Firstly, a case–control design was adopted in this study. Consequently, this design can only suggest the possible influencing factors for MCI and cannot clarify the causal relationships. Moreover, it is difficult to determine whether a poor diet leads to MCI or MCI results in a poor diet. Secondly, the information, such as dietary data, was self-reported and might have been influenced by recall bias. Thirdly, the cognitive level status may vary at different periods and under different interventions. Since this study only selected a single sample, it cannot objectively present the dynamic change process. Future studies can collect the results of multiple blood samples as well as information on interventions. Finally, considering that the surveyed regions in this study had a relatively good economic situation, the epidemic characteristics of diseases and the health awareness of the population were different from those in less developed areas. Meanwhile, the coverage areas and sample sizes of this study are limited, so the generalizability of the results requires further verification. Therefore, it is still necessary to conduct larger-scale studies in the future.

## 5. Conclusions

To summarize, this study indicates potential sex disparities in the risk factors for MCI. Women who are not in marital status, or depressed, have a low uPDI score, or have a low intake of grains, along with men living in rural areas or having a relatively low intake of grains, vegetables, and vegetable oils, constitute the key populations that merit particular attention regarding MCI. Future research should prospectively establish causal relationships. Additionally, precise intervention strategies are urgently needed.

## Figures and Tables

**Figure 1 nutrients-17-00248-f001:**
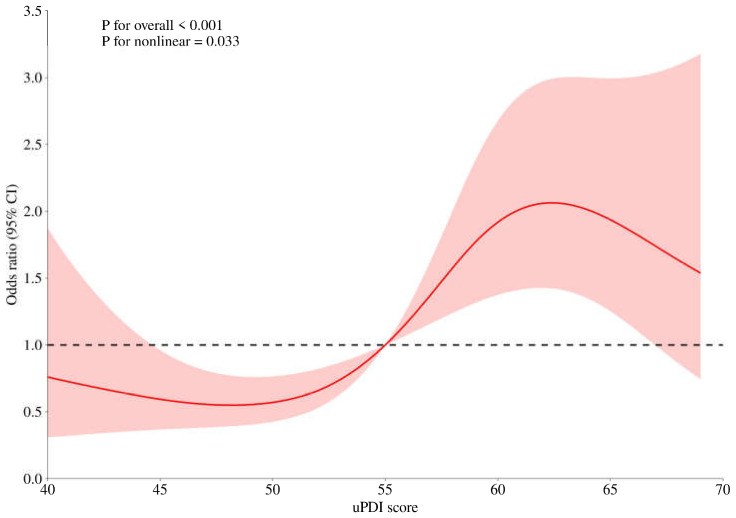
Relationship between continuous changes in uPDI and the occurrence of MCI among female elder population based on the RCS model. The red area represents the confidence interval.

**Table 1 nutrients-17-00248-t001:** Main characteristics of study participants stratified by sex.

Variables			Women				Men			
			MCI, *n* (%)	Normal, *n* (%)	χ^2^	*p*	MCI, *n* (%)	Normal, *n* (%)	χ^2^	*p*
Total			141 (24.7)	430 (75.3)			126 (24.5)	388 (75.5)		
Region					4.41	0.04			0.11	0.74
		Urban	90 (63.8)	231 (53.7)			59 (46.8)	175 (45.1)		
		Rural	51 (36.2)	199 (46.3)			67 (53.2)	213 (54.9)		
Age, years					19.48	<0.001			24.59	<0.0001
		55~65	37 (26.2)	166 (38.6)			26 (20.6)	143 (36.9)		
		65~75	58 (41.1)	195 (45.4)			52 (41.3)	175 (45.1)		
		>75	46 (32.6)	69 (16.1)			48 (38.1)	70 (18.0)		
Educational level					1.42	0.23			3.28	0.07
		Junior school or below	126 (91.3)	374 (87.6)			110 (89.4)	318 (82.6)		
		High school or above	12 (8.7)	53 (12.4)			13 (10.6)	67 (17.4)		
Occupation					0.44	0.51			0.92	0.34
		Employed or re-employ after retirement or seeking employment	127 (90.1)	395 (91.9)			101 (80.2)	295 (76.0)		
		Retirement or unemployed	14 (9.9)	35 (8.1)			25 (19.8)	93 (24.0)		
Marital status					4.62	0.03			1.44	0.23
		Married	104 (73.7)	353 (82.1)			117 (92.9)	346 (89.2)		
		Unmarried/divorced/widowed	37 (26.2)	77 (17.9)			9 (7.1)	42 (10.8)		
BMI (kg/m^2^)					0.25	0.88			2.23	0.33
		<24.0	74 (54.8)	241 (57.0)			73 (59.4)	194 (52.2)		
		≥24~<28	45 (33.3)	137 (32.4)			43 (35.0)	147 (39.5)		
		≥28	16 (11.9)	45 (10.6)			7 (5.7)	31 (8.3)		
Live alone					3.11	0.08			0.35	0.55
		Yes	128 (90.8)	408 (94.9)			122 (96.8)	371 (95.6)		
		No	13 (9.2)	22 (5.1)			4 (3.2)	17 (4.4)		
Depression					20.66	<0.001			3.34	0.07
		Yes	127 (90.1)	423 (98.4)			117 (92.9)	375 (96.7)		
		No	14 (9.9)	7 (1.6)			9 (7.1)	13 (3.4)		
Cereal ^#^					16.69	<0.001			25.16	<0.0001
		<300 g/d	94 (66.5)	201 (46.9)			75 (59.5)	133 (34.3)		
		≥300 g/d	47 (33.3)	228 (53.2)			51 (40.5)	255 (65.7)		
Vegetables intake					8.00	0.005			10.37	0.001
		<150 g/d	89 (63.1)	212 (49.4)			72 (57.1)	158 (40.7)		
		≥150 g/d	52 (36.9)	217 (50.6)			54 (42.9)	230 (59.3)		
Fruit intake					9.03	0.003			6.12	0.01
		<50 g/d	95 (67.4)	227 (52.9)			92 (73.0)	236 (60.8)		
		≥50 g/d	46 (32.6)	202 (47.1)			34 (27.0)	152 (39.2)		
Soybean intake					4.15	0.004			1.50	0.002
		<38 g/d	106 (75.2)	283 (66.0)			84 (66.7)	235 (60.6)		
		≥38 g/d	35 (24.8)	146 (34.0)			42 (33.3)	153 (39.4)		
Vegetable oil intake					6.83	0.009			6.61	0.01
		<22 g/d	93 (66.0)	229 (53.4)			84 (66.7)	208 (53.6)		
		≥22 g/d	48 (34.0)	200 (46.6)			42 (33.3)	180 (46.4)		
Strong tea intake					0.39	0.54			0.35	0.56
		no	130 (91.2)	389 (90.5)			94 (74.6)	279 (71.9)		
		yes	11 (7.8)	41 (9.5)			32 (25.4)	109 (28.1)		
Nuts intake					6.76	0.009			2.73	0.10
		<5 g/d	109 (77.3)	282 (65.6)			94 (74.6)	259 (66.8)		
		≥5 g/d	32 (22.7)	148 (34.4)			32 (25.4)	129 (33.3)		
Plant-based diet indices *		PDI	48.2 ± 6.4	49.7 ± 6.5	2.10	0.04	48.6 ± 5.9	50.1 ± 6.7	2.11	0.35
		hPDI	56.4 ± 5.6	57.6 ± 4.8	2.16	0.03	57.3 ± 4.8	57.4 ± 5.0	0.19	0.85
		uPDI	57.7 ± 7.4	54.7 ± 7.6	−3.78	0.0002	56.32 ± 7.4	54.02 ± 7.5	−2.77	0.006
Sarcopenia related										
	Sarcopenia				0.34	0.56			1.65	0.20
		No	134 (97.1)	410 (96.0)			113 (90.4)	362 (93.8)		
		Yes	4 (2.9)	17 (4.0)			12 (9.6)	24 (6.2)		
	Low muscle mass				1.35	0.25			1.35	0.25
		No	122 (95.3)	347 (92.3)			92 (85.2)	285 (89.3)		
		Yes	6 (4.7)	29 (7.7)			16 (14.8)	34 (10.7)		
	Low muscle strength or performance				7.30	0.03			4.20	0.12
		No low muscle strength or performance	69 (50.4)	216 (50.8)			49 (39.5)	158 (41.9)		
		Low muscle strength or performance	48 (35.0)	178 (41.9)			50 (40.3)	171 (45.4)		
		Low muscle strength and performance	20 (14.6)	31 (7.3)			25 (20.2)	48 (12.7)		

* Values are mean ± SD; ^#^ cereal included rice and wheat and their product.

**Table 2 nutrients-17-00248-t002:** Analysis of the risk of MCI in elderly women aged 55 and above by multivariate logistic regression model.

Variables		β	$s\bar{x}$	Wald χ^2^	OR (95% CI)	*p*
Intercept		−3.97	0.90	19.5336		<0.0001
Marital status						
	Married					
	Unmarried/divorced/widowed	0.67	0.29	5.27	1.95 (1.10~3.45)	0.02
Depression						
	Yes	1.80	0.60	9.01	6.06 (1.87~19.66)	0.003
	No					
Cereal						
	<300 g/d					
	≥300 g/d	−1.15	0.26	19.97	0.32 (0.19~0.53)	<0.0001
uPDI		0.05	0.02	12.21	1.06 (1.02~1.09)	0.0005

**Table 3 nutrients-17-00248-t003:** Analysis of the risk of MCI in elderly men aged 55 and above by multivariate logistic regression model.

Variables		β	$s\bar{x}$	Wald χ^2^	OR (95% CI)	*p*
Intercept		0.90	0.31	8.40		0.004
Region						
	Urban					
	Rural	0.64	0.27	5.57	1.90 (1.12~3.25)	0.02
Age, years						
	55~65					
	65~75	0.673	0.31	4.83	1.96 (1.08~3.57)	0.03
	>75	1.55	0.34	21.22	4.71 (2.44~9.12)	<0.0001
Vegetables intake						
	<150 g/d					
	≥150 g/d	−0.94	0.27	12.48	0.39 (0.23~0.66)	0.0004
Vegetable oil intake						
	<22 g/d					
	≥22 g/d	−0.69	0.25	7.55	0.50 (0.31~0.82)	0.006
Cereal						
	<300 g/d					
	≥300 g/d	−0.83	0.25	11.29	0.44 (0.27~0.71)	0.0008

## Data Availability

The data presented in this study are available on request from the corresponding author. The data are not publicly available due to privacy and ethical concerns in this survey.

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
