# Peer review of "Sex Disparity in the Nutrition-Related Determinants of Mild Cognitive Impairment: A Case–Control Study"

_nutrients, 2025, doi:10.3390/nu17020248_

Round 1

Reviewer 1 Report

Comments and Suggestions for Authors

The objective of this research was to evaluate the factors related to mild cognitive impairment in men and women (separately) in an older adult population. The study was a nested case-control study. The authors provided sufficient rationale to conduct this work using the appropriate methods. My comments:

Title: In the title and throughout the manuscript “gender” may be more appropriately replaced with “sex”.  Sex is typically assigned at birth. gender refers to socially constructed roles, expectations and behaviors.

Abstract:

  • Line 15: if the PID has been validated for the use in this study please indicate/

  • Line 21: please indicate the intakes are in grams per day. Do this as well throughout the manuscript.

  • Lines 30-31: Please indicate why precise intervention strategies are urgently needed. For example, “Our research indicates that sex disparities warrant future studies to evaluate preventive intervention strategies for MCI.”

Materials and Methods:

  • Lines 143-144: This sentence is not clear. Is this what is meant? “Eighteen types of plant-based foods from the 81 categories were extracted from the FFQ.”

  • Line 146: the PDI was based on U.S. dietary data. Was the PDI used in the present study validated with Chinese food data?

  • Line 146: It would be useful to know the range of values for PDI.

Results: 

  • Line 216: Please indicate that the association was positive.

  • Table 1: Age, year should be Age, years

  • Table 1: “Strong teat” should be “Strong tea”. Also, please define in the text what is meant by strong tea.

Discussion:

  • Line 245: Please change to “early warning signs”

  • Line 247: Please edit to “... to explore the risk factors for MCI in men and women e\separately.”

  • Line 260: I do not see in the Methods that the MMSE was given. This paragraph seems more appropriate in the Methods section.

  • Line 338: Please change “two genders” to “men and women”

  • Please think about the possibility of reverse causation. Rather than diet being related to MCI, MCI is related to poor diet.

  • Please consider a discussion on the translatability to other geographies.

Summary & Conclusions:

  • The summary and conclusions are adequately discussed.

Author Response

Comments 1:

Title: In the title and throughout the manuscript “gender” may be more appropriately replaced with “sex”.  Sex is typically assigned at birth. gender refers to socially constructed roles, expectations and behaviors.

Response: In the title and throughout the manuscript, “gender” has replaced with “sex”.

Comments 2:

Line 15: if the PID has been validated for the use in this study please indicate.

Response: The calculation method of the PDI (Plant-based Diet Index) directly adopts the plant-based dietary pattern established by Ambika Satija in Harvard.

Comments 3:

Line 21: please indicate the intakes are in grams per day. Do this as well throughout the manuscript.

Response: It has been indicated that intakes are in grams per day throughout the manuscript.

Comments 4:

Lines 30-31: Please indicate why precise intervention strategies are urgently needed. For example, “Our research indicates that sex disparities warrant future studies to evaluate preventive intervention strategies for MCI.”

Response: To date, there is no effective cure available for delaying the progression of AD. Therefore, primary prevention of AD is of crucial importance. Identifying high-risk groups can maximize early warning and precise intervention.

Comments 5:

Lines 143-144: This sentence is not clear. Is this what is meant? “Eighteen types of plant-based foods from the 81 categories were extracted from the FFQ.”

Response: Yes. It is meant “Eighteen types of plant-based foods from the 81 categories were extracted from the FFQ.”, and it has been revised in the manuscript.

Comments 6:

Line 146: the PDI was based on U.S. dietary data. Was the PDI used in the present study validated with Chinese food data?

Response: In this study the PDI used in the present study did not validate with Chinese food data.

Comments 7:

Line 146: It would be useful to know the range of values for PDI.

Response: The range of values for PDI has been added to the supplementary materials.

Comments 8:

Line 216: Please indicate that the association was positive.

Response: It has been revised in the text.

Comments 9:

Table 1: Age, year should be Age, years

Response: It has been revised in Table 1 and Table 3.

Comments 10:

Table 1: “Strong teat” should be “Strong tea”. Also, please define in the text what is meant by strong tea.

Response: “Strong teat” had been revised to “Strong tea”, and A strong tea was defined as the ratio of tea leaves to water at 1:50 in the text. 

Comments 11:

Line 245: Please change to “early warning signs”

Response: It has been revised in the text.

Comments 12:

Line 247: Please edit to “... to explore the risk factors for MCI in men and women e\separately.”

Response: It has been revised in the text.

Comments 13:

Line 260: I do not see in the Methods that the MMSE was given. This paragraph seems more appropriate in the Methods section.

Response: It has been revised in the text according to the comment.

Comments 14:

Line 338: Please change “two genders” to “men and women”

Response: It has been revised in the text.

Comments 15:

Please think about the possibility of reverse causation. Rather than diet being related to MCI, MCI is related to poor diet.

Response: This suggestion experts are very good. This part of the content has been already added to the section of deficiencies in the discussion.

Comments 16:

Please consider a discussion on the translatability to other geographies.

Response: It has been revised in the discussion.

Reviewer 2 Report

Comments and Suggestions for Authors

In the current study the authors explored the nutrition-related influencing factors of mild cognitive impairment between males and females from 4 sites in Zhejiang Province, China in 2020 by using the method of nested case control study, thereby achieving precise prevention on mild cognitive impairment. The study indicates potential gender disparities in the risk factors for mild cognitive impairment.

Some suggestions:

1. Introduction, page 1: please give statistical data relating to our days, not to the years 2020-21

2. page 3, Point 2.2. Sampling Method and Study Population:

- lines 108-109, you wrote as inclusion criteria “patients free from comorbid conditions that could interfere with the assessment, such as congenital  or acquired mental retardation, and uncorrectable visual or hearing abnormalities”. Weren't patients with other diseases, with certain medications excluded from the study? Please clarify.

- Line 116, you wrote: “Ultimately, a total of 1086 participants were included in the analysis”. Please add how many patients you had at the beginning and specify the number of excluded patients and the reason (for each reason the No of patients excluded). For these explanations I recommend you to add a figure.

3. Please add as supplementary material the Questionnaire used for the patients.

4. Give please more details concerning the following statements:

- “The specific cultural and linguistic adaptations made to the original English version to develop the MoCA Beijing version were also described in detail.” (Point 2.4 Cognitive function assessment , lines 131-133).

- “Sarcopenia, low muscle mass, low muscle strength and low performance were defined based on the recommendations of the Asian Working Group for Sarcopenia 2019 (AWGS2019) consensus.” (Point 2.6. Assessment of Covariates, lines 173-75).

5. The amount of vegetables, cereals consumed was expressed in g. Is g/day?

6. Discussion, page 11: I don’t find the reason to write about:

- the caffeine and chocolate consumption as long as these aspects were not followed in your study (lines 272-79).

- sweet foods as log as the consumption of sweet food among Chinese elderly  people is very low (line 291-92).

Author Response

Comments 1:

Introduction, page 1: please give statistical data relating to our days, not to the years 2020-21

Response: It has been revised in the introduction.

Comments 2:

page 3, Point 2.2. Sampling Method and Study Population:

- lines 108-109, you wrote as inclusion criteria “patients free from comorbid conditions that could interfere with the assessment, such as congenital  or acquired mental retardation, and uncorrectable visual or hearing abnormalities”. Weren't patients with other diseases, with certain medications excluded from the study? Please clarify.

- Line 116, you wrote: “Ultimately, a total of 1086 participants were included in the analysis”. Please add how many patients you had at the beginning and specify the number of excluded patients and the reason (for each reason the No of patients excluded). For these explanations I recommend you to add a figure.

Response: Since this study only focuses on exploring the risk factors for three diseases, (epilepsy, Parkinson's disease, and Alzheimer's disease), which require screening with scales, subjects with certain diseases that may affect the assessment were excluded. These mainly include congenital or acquired mental retardation, and uncorrectable visual or hearing abnormalities. Other conditions were not excluded.

At the beginning of the study, 2,790 subjects were included in the survey. Subjects with complete data regarding basic and health, cognitive examination, psychological evaluation, physical examinations and a survey of basic daily living abilities were selected to participate in the present study( n=1103). Based on the definition of MCI, subjects were excluded if they were unable to carry out basic activities of daily living, including eating, dressing, bathing, using the toilet, grooming, transferring between the bed and chair, walking across a room, and maintaining urinary or fecal continence (n=17). Regarding participants whose sleep duration fell below the 1st percentile (P1) or above the 99th percentile (P99) in each age group's sleep duration distribution, we replaced these extreme values with the corresponding P1 and P99 values, respectively. Ulti-mately, a total of 1086 participants were included in the analysis. Above have been added to 2.2. Sampling Method and Study Population. Since the screening process is relatively straightforward, I suggest that there is no need to add a figure.

Comments 3:

 Please add as supplementary material the Questionnaire used for the patients.

Response: Regarding the development of the scale questionnaire, due to confidentiality concerns, I’m sorry that it is not convenient to add the questionnaire used for the patients.

Comments 4:

Give please more details concerning the following statements:

- “The specific cultural and linguistic adaptations made to the original English version to develop the MoCA Beijing version were also described in detail.” (Point 2.4 Cognitive function assessment , lines 131-133).

- “Sarcopenia, low muscle mass, low muscle strength and low performance were defined based on the recommendations of the Asian Working Group for Sarcopenia 2019 (AWGS2019) consensus.” (Point 2.6. Assessment of Covariates, lines 173-75).

Response: The details have been added to the text.

Comments 5:

The amount of vegetables, cereals consumed was expressed in g. Is g/day?

Response: It has been revised in the text.

Comments 6:

Discussion, page 11: I don’t find the reason to write about:

- the caffeine and chocolate consumption as long as these aspects were not followed in your study (lines 272-79).

- sweet foods as log as the consumption of sweet food among Chinese elderly  people is very low (line 291-92).

Response: The reason is that the data surveyed in this study shows that the number of consumers and the consumption volume of coffee, chocolate, and other sweets are very low. See the table below for details. This table has been already added to the supplementary materials.
